# Systematic comparison of respiratory syncytial virus-induced memory B cell responses in two anatomical compartments

Laila Shehata[1], Wendy F. Wieland-Alter[2], Daniel P. Maurer[1], Eunice Chen ⓘ [3], Ruth I. Connor[2], Peter F. Wright[2] & Laura M. Walker[1]

Respiratory syncytial virus (RSV) is a leading cause of hospitalization in infants and young children. Although it is widely agreed that an RSV vaccine should induce both mucosal and systemic antibody responses, little is known about the B cell response to RSV in mucosa-associated lymphoid tissues. Here, we analyze this response by isolating 806 RSV F-specific antibodies from paired adenoid and peripheral blood samples from 4 young children. Overall, the adenoid-derived antibodies show higher binding affinities and neutralization potencies compared to antibodies isolated from peripheral blood. Approximately 25% of the neutralizing antibodies isolated from adenoids originate from a unique population of IgM+ and/or IgD+ memory B cells that contain a high load of somatic mutations but lack expression of classical memory B cell markers. Altogether, the results provide insight into the local B cell response to RSV and have implications for the development of vaccines that stimulate potent mucosal responses.

[1] Adimab LLC, Lebanon, NH 03766, USA. [2] Department of Pediatrics, Geisel School of Medicine at Dartmouth, Hanover, NH 03755, USA. [3] Department of Surgery, Geisel School of Medicine at Dartmouth, Hanover, NH 03755, USA. Correspondence and requests for materials should be addressed to L.M.W. (email: laura.walker@adimab.com)

Respiratory syncytial virus (RSV) causes substantial morbidity and mortality in infants and young children, and there are currently no licensed vaccines to protect these high-risk populations[1]. There are several barriers to the development of an RSV vaccine, including the young age at which primary infection occurs, the legacy of vaccine-enhanced disease, and the lack of animal models that fully recapitulate the pathogenesis of RSV infection in humans[2,3]. Although there are no clinically approved RSV vaccines, there are 43 vaccine candidates in development, of which 19 are in clinical stage development[4]. Most of these vaccines seek to induce neutralizing antibodies that recognize the RSV fusion (F) glycoprotein, which is targeted by the prophylactic antibody palivizumab and the majority of RSV-specific neutralizing antibodies in human sera[5–8].

RSV F is a class I fusion protein that mediates viral entry by transitioning from a metastable prefusion conformation (preF) to a highly stable postfusion (postF) conformation[9]. Over the past several years, epitope mapping studies using both human and murine monoclonal antibodies have defined at least 6 major antigenic sites on the RSV F protein[2,5,10–13]. Some of these sites are expressed on both preF and postF, while other antigenic sites are preferentially or exclusively presented on only one conformation. Importantly, multiple recent studies have shown that the vast majority of highly potent neutralizing antibodies to RSV target preF-specific epitopes[5–7,14]. Hence, vaccines that preserve preF-specific antigenic surfaces may have great clinical potential.

RSV replicates exclusively in respiratory epithelial cells, initiating infection in the upper respiratory tract and in some cases progressing to the lower respiratory tract. Thus, it is widely believed that an ideal RSV vaccine should induce systemic and mucosal immune responses that protect both the upper and lower respiratory tracts[15]. Importantly, a substantial body of literature suggests that RSV-specific mucosal antibody levels correlate more strongly with protection against RSV infection than serum antibody titers[16–22]. For example, a recent clinical study in a pediatric cohort showed that high levels of RSV-specific mucosal IgG correlated with reduced viral load and inflammation, whereas plasma IgG levels were not predictive of either[17]. In addition, experimental RSV-challenge studies in adult donors have shown that nasal antibody titers correlate with protection from RSV infection[19]. Finally, preclinical immunogenicity and efficacy studies utilizing a live-attenuated vaccine candidate, RGΔM2-2, showed that the protective efficacy of this vaccine was significantly higher when delivered by the intranasal route compared to the intramuscular route, despite both immunizations inducing comparable serum antibody titers[23]. Although these studies provide compelling evidence that mucosal immunity will be required for efficient protection against RSV, little is known about the anatomic location(s) of RSV-specific memory B cells within mucosa-associated lymphoid tissues, the specificities and functional properties of these antibodies, and if/how the RSV-specific mucosal antibody response differs from the systemic antibody response.

To address these questions, we isolated and characterized over 800 RSV F-specific antibodies from paired peripheral blood and adenoid tissues obtained from 4 young children undergoing adenoidectomy. RSV F-specific memory B cells were present in the adenoids of all children, and in most donors, a higher proportion of adenoid-derived antibodies showed neutralizing activity compared to the corresponding peripheral blood mononuclear cell (PBMC)-derived antibodies. Furthermore, a relatively large fraction of the adenoid-derived neutralizing antibodies originated from a unique population of memory B cells that were not class-switched and lacked expression of classical memory B cell markers. Importantly, nearly all the highly potent neutralizing antibodies isolated from both compartments targeted epitopes exclusively expressed on preF. Taken together, our results demonstrate that natural RSV infection induces robust memory B cell responses in the adenoids of young children and provide strong rationale for the development of preF-based mucosal vaccines that boost local neutralizing responses.

## Results

**Isolation of RSV F-specific B cells from adenoid and blood.** To analyze and compare the memory B cell response to natural RSV infection in adenoids and peripheral blood, we obtained paired adenoid tissue and peripheral blood samples from 6 young children, aged 2.5–4 years, who were undergoing adenoidectomy for clinical conditions unrelated to RSV infection (Supplementary Table 1). Adenoids were used as a representative source of respiratory mucosal lymphocytes because this lymphoid tissue has been previously shown to be an important induction site for B cells that migrate to the respiratory tract and associated glands[24–26]. The anatomical location of the adenoids—at the site of entry into the upper respiratory tract—also suggests a potential role in anti-RSV immunity. Although none of the children had a documented history of RSV infection, previous studies have shown that essentially all children have been infected by RSV at least once by the age of 2[27]. Consistent with the notion of prior RSV exposure, plasma samples obtained from all 6 children displayed neutralizing activity against RSV-A2 (Supplementary Table 1). Additionally, samples collected directly from the mucosal surface of the excised tissues using absorbent filter paper and a sterile wash solution had measurable RSV neutralizing activity, suggesting the presence of neutralizing antibodies in the mucosal secretions (Supplementary Table 1).

To assess the magnitude of the RSV F-specific B cell response in both anatomical compartments, the adenoid and PBMC samples were stained with a panel of B cell surface markers (CD19, CD20, IgG, IgA, CD27, and FcRL4) and fluorescently labeled tetramers of RSV pre-F and post-F, and analyzed by flow cytometry (Fig. 1a, Supplementary Fig. 1). RSV F-reactive B cells were detected in the adenoid samples from all 6 donors but were only observed in 4 of the 6 corresponding PBMC samples (Fig. 1b, Supplementary Fig. 1). The frequency of RSV F-specific B cells among total CD19[+] B cells in the adenoid and PBMC samples ranged from 0.03 to 0.22 and from 0 to 0.19%, respectively (Fig. 1b, Supplementary Fig. 1). Notably, we observed no correlation between (i) the frequency of RSV F-reactive B cells in peripheral blood and adenoid tissue and (ii) plasma neutralization titer and the frequency of RSV F-reactive B cells in either compartment (Supplementary Fig. 2). The latter result is consistent with previous studies showing a lack of correlation between the frequencies of antigen-specific memory B cells and serum titers of antigen-specific IgG[28].

We next single-cell sorted between 100–300 RSV F-reactive B cells from both the adenoid and PBMC samples from each of the 4 donors that had detectable RSV F-specific B cell responses in both compartments. Although we sorted all B cells that were RSV F-reactive (regardless of B cell phenotype), we used index sorting to track the B cell surface markers expressed on each sorted cell. This analysis showed that the RSV F-specific B cell subset distribution varied considerably both between the adenoid and peripheral blood compartments and among the 4 donors (Fig. 1c, d). For example, in some donors, there was a higher proportion of RSV F-specific IgG[+] memory B cells in peripheral blood compared to adenoid tissue (e.g., donor 2635 and donor 2849), whereas the converse was observed in other donors (e.g., donor 2665 and donor 2666) (Fig. 1c, d). Notably, in all 4 donors, there was little to no enrichment for RSV F-specific IgA[+] B cells

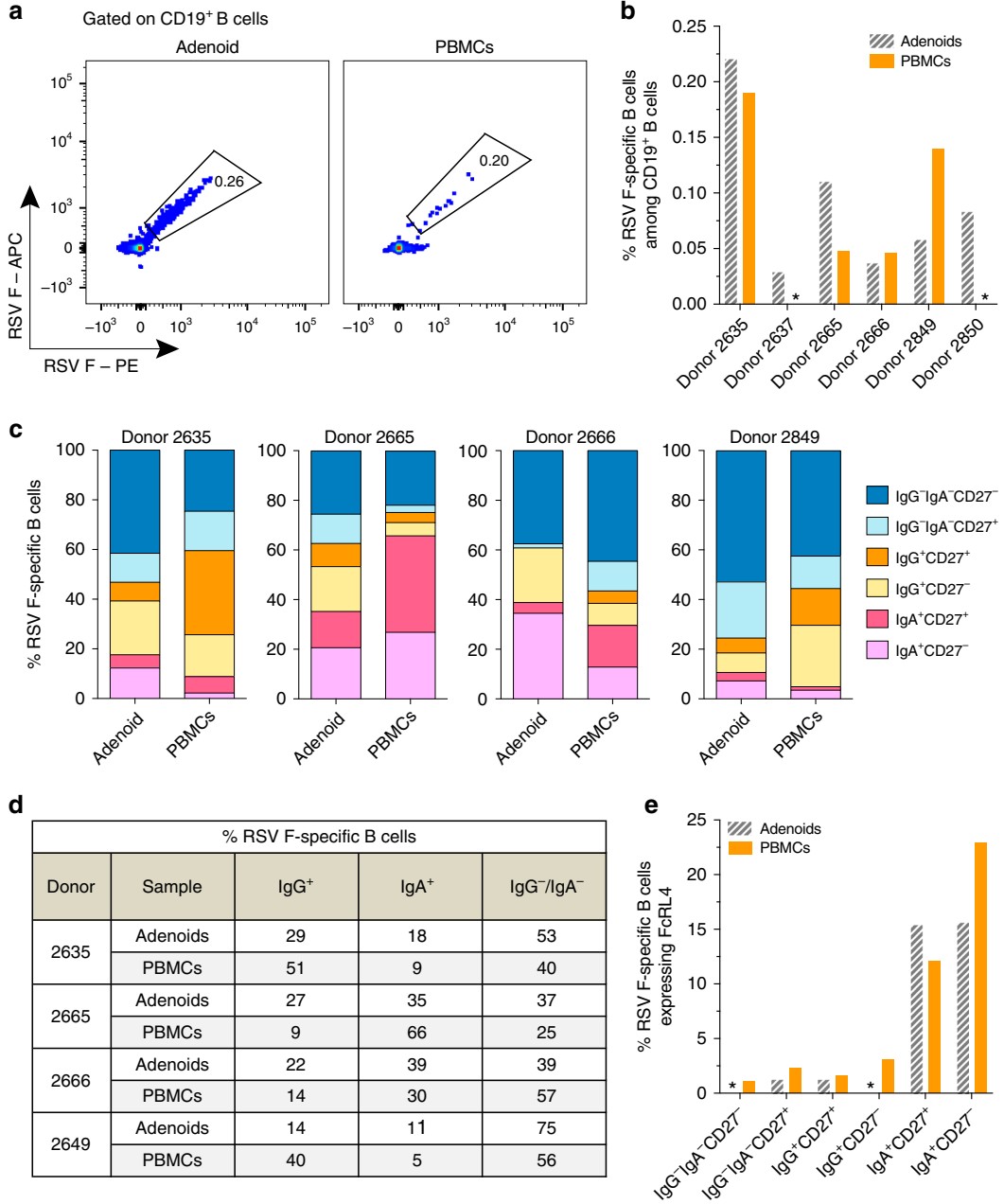

**Fig. 1** RSV F-specific B cell responses in the adenoids and peripheral blood of young children. Shown is the frequency of RSV F-specific B cells among CD19+ B cells in adenoid (left) and PBMCs (right) for a representative donor (**a**) and a summary for all 6 donors analyzed (**b**). The frequency of RSV F-reactive B cells within the CD14−CD3−CD8−CD19+ population is shown inside the gate. Asterisks in **b** indicate B cell responses that were below the limit of detection. **c** Index sort analysis of surface markers expressed on B cells from which RSV F-reactive antibodies were isolated. **d** Isotype distribution of RSV F-specific B cells in adenoid and PBMCs. **e** Percentage of RSV F-reactive, FcRL4+ B cells within each memory B cell subset described in **c**

in the adenoid samples relative to the corresponding PBMC samples, and in one donor (donor 2665) there was a substantially higher proportion of RSV F-specific IgA+ B cells in peripheral blood compared to adenoid tissue (Fig. 1c, d). Furthermore, in all 4 donors, a considerable proportion (21–52%) of RSV F-specific B cells in both compartments were not class-switched and lacked the expression of the hallmark memory B cell marker CD27. Since previous studies have shown that the inhibitory receptor FcRL4 is expressed on a subset of tissue-resident memory B cells[29], we also analyzed whether this marker was preferentially expressed on RSV F-specific adenoid B cells. Although the proportion of total B cells expressing FcRL4 was higher in

adenoid tissue compared to peripheral blood for most donors (Supplementary Fig. 3), the proportion of RSV F-reactive B cells expressing FcRL4 was similar in both compartments (Fig. 1e). In both adenoid and peripheral blood, a small proportion of RSV F-specific B cell clones within most of the memory B cell subsets expressed FcRL4, with the exception of the adenoid-resident IgG− IgA−CD27− and IgG+CD27− subsets (Fig. 1e). Consistent with previous reports, the large majority of RSV F-specific FcRL4+ B cells in both compartments were of the IgA isotype[30,31] (Fig. 1e). We conclude that the natural RSV infection induces memory B cell responses in the adenoids of young children, and that the contribution of different memory B cell

populations to the RSV F-specific response varies among donors and between the adenoid and peripheral blood compartments.

**Identification of atypical memory B cells in adenoid tissue.** To further characterize the RSV F-specific B cell responses in both compartments, we amplified the antibody variable heavy (VH) and variable light (VL) chain sequences from the sorted B cells by single cell-PCR and cloned over 800 cognate VH-VL pairs into an IgG1 expression vector for sequencing and production.

Sequence analysis revealed that the RSV F-specific antibody repertoires were highly diverse in both compartments in all 4 donors, each containing few to no expanded clonal lineages (Supplementary Fig. 4A). The median CDRH3 lengths among the PBMC-derived and adenoid-derived antibodies were 15 and 16 amino acids, respectively, which is consistent with previously reported median CDRH3 lengths for anti-viral antibodies (Supplementary Fig. 4B)[32]. Although VH germline gene usage was also comparable between the two compartments, there was an enrichment for VH5-51 and VH1-69 in the adenoid-derived antibody panel and an enrichment for VH4-34 and VH3-30 in the PBMC-derived antibody panel (Supplementary Fig. 4C). The load of somatic mutations varied among the 4 donor repertoires, with the median number of VH nucleotide substitutions ranging from 8 to 11 in the adenoid-derived antibodies and 7–9 in the PBMC-derived antibodies (Fig. 2a, Supplementary Fig. 5). This level of somatic hypermutation (SHM) is similar to that observed in RSV F-specific antibodies isolated from infants following primary RSV exposure and significantly lower than that observed in RSV F-specific antibodies isolated from healthy adult donors, suggesting that these young children have likely experienced a limited number of RSV infections[5,14]. Notably, for 3 out of 4 donors, the overall level of SHM trended higher in

the adenoid-derived antibodies relative to the PBMC-derived antibodies, but this difference only reached statistical significance in donor 2665.

Interestingly, analysis of SHM load clustered by memory B cell subset revealed that the vast majority (88%) of antibodies derived from IgG⁻IgA⁻CD27⁻ adenoid B cells contained a high load of somatic mutations, providing evidence for antigen-mediated selection in germinal centers (Fig. 2b). These antibodies also showed an enrichment for replacement mutations over silent mutations in complementary determining regions (CDRs) 1 and 2 compared with framework regions (FWRs) 1–3, which is a hallmark of affinity maturation (Supplementary Fig. 6). Unexpectedly, the average level of SHM in these antibodies was similar to those derived from classical IgG⁺CD27⁺ and IgA⁺CD27⁺ memory B cells, suggesting similar antigenic selection characteristics (Fig. 2b). In contrast, the majority of antibodies (68%) derived from IgG⁻IgA⁻CD27⁻ peripheral blood B cells lacked SHM, suggesting a naïve B cell origin (Fig. 2b). In all 4 donors, the percentage of RSV F-specific antibodies isolated from atypical memory B cells (defined as IgG⁻IgA⁻CD27⁻ B cells containing SHM) was significantly higher in adenoids compared to PBMCs (Fig. 2c).

To further characterize this atypical memory B cell subset, we analyzed the IgM, IgD, CD5, and CD45RB^MEM55 expression profiles of RSV F-specific adenoid B cells that lacked expression of CD27. We included CD5 and CD45RB^MEM55 in the staining panel because previous studies have shown that a sizable fraction of tonsillar IgM⁺IgD⁺CD38⁻CD27⁻CD45RB^MEM55+ and CD19⁺CD5⁺ B cells express somatically mutated VH and VL variable regions[33–36]. This analysis revealed that (i) the large majority of B cells within the RSV F-specific IgG⁻IgA⁻CD27⁻ subset either co-expressed IgM and IgD or only expressed IgM (Fig. 2d) and (ii) the vast majority (85–98%) of these IgM⁺ and/

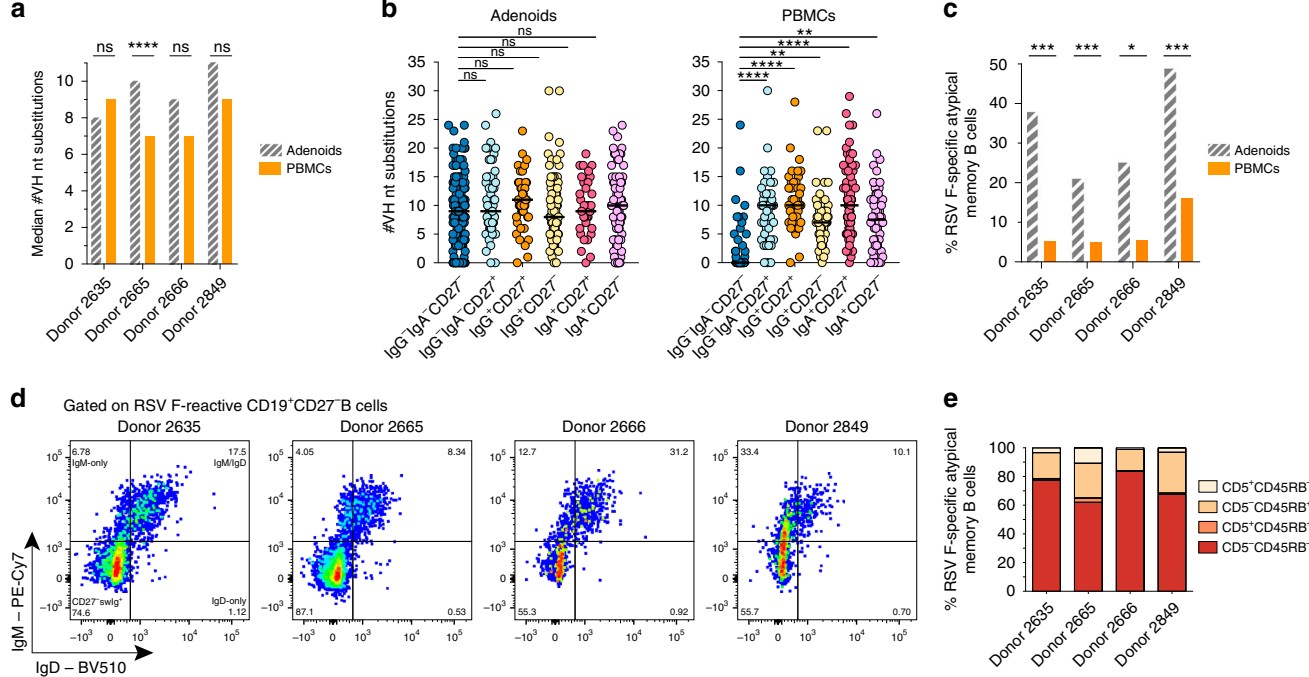

**Fig. 2** Sequence characteristics of RSV F-specific antibodies. **a** Median number of VH nucleotide substitutions in RSV F-specific antibodies isolated from adenoids or PBMCs. **b** Number of VH nucleotide substitutions in antibodies isolated from each B cell subset. Each point represents an individual antibody. Black bars indicate median values. **c** Percentage of RSV F-specific atypical memory B cells (defined as CD19⁺IgG⁻IgA⁻CD27⁻ and containing SHM) in adenoids and peripheral blood. **d** Percentage of RSV F-specific CD19⁺CD27⁻ B cells in the adenoids of each donor that are class-switched or express IgM and/or IgD. **e** Percentage of CD19⁺RSV-F⁺CD27⁻IgM⁺/IgD⁺ adenoid B cells expressing CD5 or CD45RB^MEM55. Statistical comparisons were made using the Mann-Whitney test and Fisher's exact test (****$P < 0.0001$, ***$P < 0.001$, **$P < 0.01$, *$P < 0.05$, n.s. not significant). Nt, nucleotide; swIg, switched Ig

or IgD$^+$ B cells either lacked expression of both CD5 and CD45RB$^{MEM55}$ (62–83%) or displayed a CD5$^-$CD45RB$^{MEM55+}$ phenotype (15–28%) (Fig. 2e, Supplementary Fig. 7). Hence, these B cells appear to belong to multiple sub-populations of IgM$^+$ and/or IgD$^+$ memory B cells that lack expression of the hallmark memory B cell marker CD27 and are heterogenous with respect to CD5 and CD45RB$^{MEM55}$ expression.

**Binding and functional analysis of RSV antibodies.** To characterize the binding properties of the isolated antibodies, we measured their apparent binding affinities ($K_D^{App}$) for RSV pre-F and post-F using biolayer interferometry. The percentage of antibodies that bound exclusively to either pre-F or post-F varied among the 4 donors but was similar between the two compartments within individual donors, with the exception of donor 2849 (Fig. 3a, Supplementary Data 1). In this donor, pre-F-specific antibodies were present at higher frequency in PBMCs compared to adenoid tissue (Fig. 3a). As observed in previous studies, a substantially larger proportion of antibodies bound exclusively to pre-F (16–65%) than to post-F (0–15%), demonstrating that the unique surfaces on pre-F are likely more immunogenic than those on post-F[5,14]. Although finer epitope mapping would be required to better resolve the differences in epitope distribution between the two compartments, these results suggest that the relative immunogenicity of the different antigenic

surfaces on RSV F is probably more dependent on the donor repertoire and/or immune history than on the anatomical site of B cell activation.

In 3 out of 4 donors, a higher proportion of RSV F-specific antibodies isolated from adenoid tissue bound with medium to high affinity to pre-F ($K_D^{App} < 5.0$ nM) compared to antibodies derived from PBMCs (Fig. 3b, Supplementary Data 1). For example, 70% of the adenoid-derived antibodies cloned from donor 2666 displayed medium to high-binding affinity to pre-F compared to only about 30% of the PBMC-derived antibodies. Similarly, for donor 2665, 60% and 6% of adenoid-derived and PBMC-derived antibodies, respectively, bound to pre-F with medium to high affinity. The above results, combined with the observation that 2 additional donors had detectable RSV F-specific B cell responses in adenoid tissue but not in peripheral blood (Fig. 1b), suggests that the B cell response to RSV in the adenoid may be more robust than the corresponding peripheral blood response.

We next tested the antibodies for neutralizing activity against RSV-A2 using a previously described microtiter assay[37]. Fourteen to 36% of the adenoid-derived antibodies and 0 to 26% of the PBMC-derived antibodies showed detectable neutralizing activity (IC$_{50}$ ≤ 25 μg/mL) in this assay (Fig. 4a, Supplementary Data 1). Notably, in all 4 donors, <20% of antibodies isolated from both compartments showed highly potent neutralizing activity (IC$_{50}$ ≤ 0.05 μg/mL), which is lower than that observed for 3 previously characterized healthy adult donors, in which 19–38% of isolated antibodies neutralized with high potency[5]. This difference is likely due to the decreased number of RSV exposures in young children compared to adult donors. Consistent with this explanation, the panel of antibodies isolated from the oldest donor (donor 2635, Supplementary Table 1) contained the highest proportion of highly potent neutralizing antibodies (Fig. 4a, Supplementary Data 1).

In agreement with the binding analysis, for 3 out of 4 donors, a larger proportion of adenoid-derived antibodies showed neutralizing activity compared to PBMC-derived antibodies (Fig. 4a). For example, for donor 2666, neutralizing antibodies were present at approximately 3 times higher frequency in adenoids compared to PBMCs. Similarly, for donor 2665, 15% of the adenoid-derived antibodies neutralized with half-maximal inhibitory concentrations (IC$_{50}$s) below 25 μg/ml, whereas none of the PBMC-derived antibodies showed detectable neutralizing activity at this concentration.

Interestingly, analysis of the relationship between memory B cell subset and neutralizing activity revealed that approximately 90% of the PBMC-derived neutralizing antibodies originated from only three B cell subsets: IgG$^+$CD27$^+$, IgG$^+$CD27$^-$, and IgG$^-$IgA$^-$CD27$^+$ (Fig. 4b). In contrast, the adenoid-derived neutralizing antibodies were more evenly distributed across the six different memory B cell populations, with the largest proportion (25%) originating from the atypical IgG$^-$IgA$^-$CD27$^-$ memory B cell subset (Fig. 4b). Of significance, the antibodies isolated from this B cell subset showed similar apparent binding affinities and neutralization potencies compared to the antibodies derived from other memory B cell subsets, providing further evidence that that these B cells are germinal center experienced (Fig. 4c, d).

Finally, we analyzed the relationship between pre-F-binding and post-F-binding activity and neutralization potency. Fifty to 60% of pre-F-specific antibodies isolated from both adenoids and PBMCs showed neutralizing activity compared to only 0–8% of post-F-specific antibodies and 10–12% of conformation-independent antibodies (Fig. 4e). Importantly, greater than 90% of highly potent antibodies (IC$_{50}$ < 0.05 μg/mL) isolated from both compartments bound exclusively to pre-F (Fig. 4f).

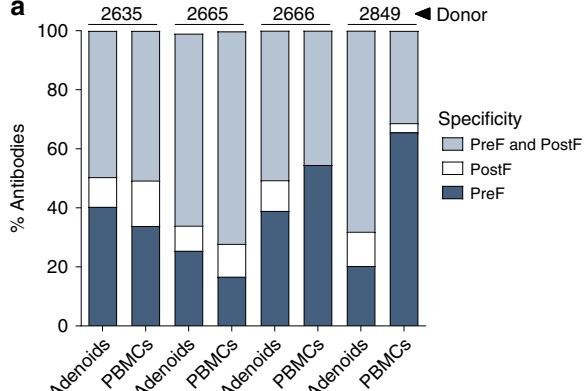

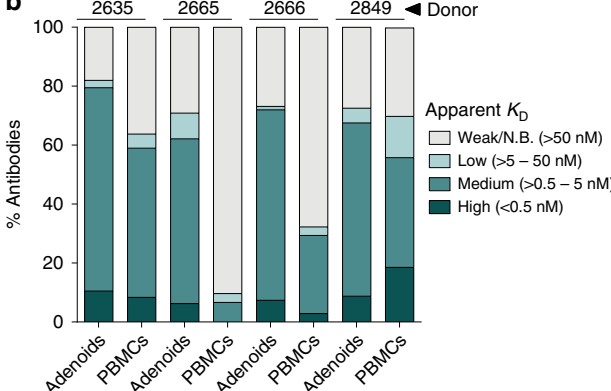

**Fig. 3** Binding properties of RSV F-specific antibodies isolated from adenoids and PBMCs. **a** Percentage of antibodies isolated from adenoids or PBMCs that are preF-specific, preF/postF-cross-reactive, or postF-specific ($n = 808$ antibodies). **b** Percentage of antibodies isolated from adenoids or PBMCs with the indicated apparent affinities for preF ($n = 808$ antibodies). N.B. non-binding. Results are representative of two independent experiments

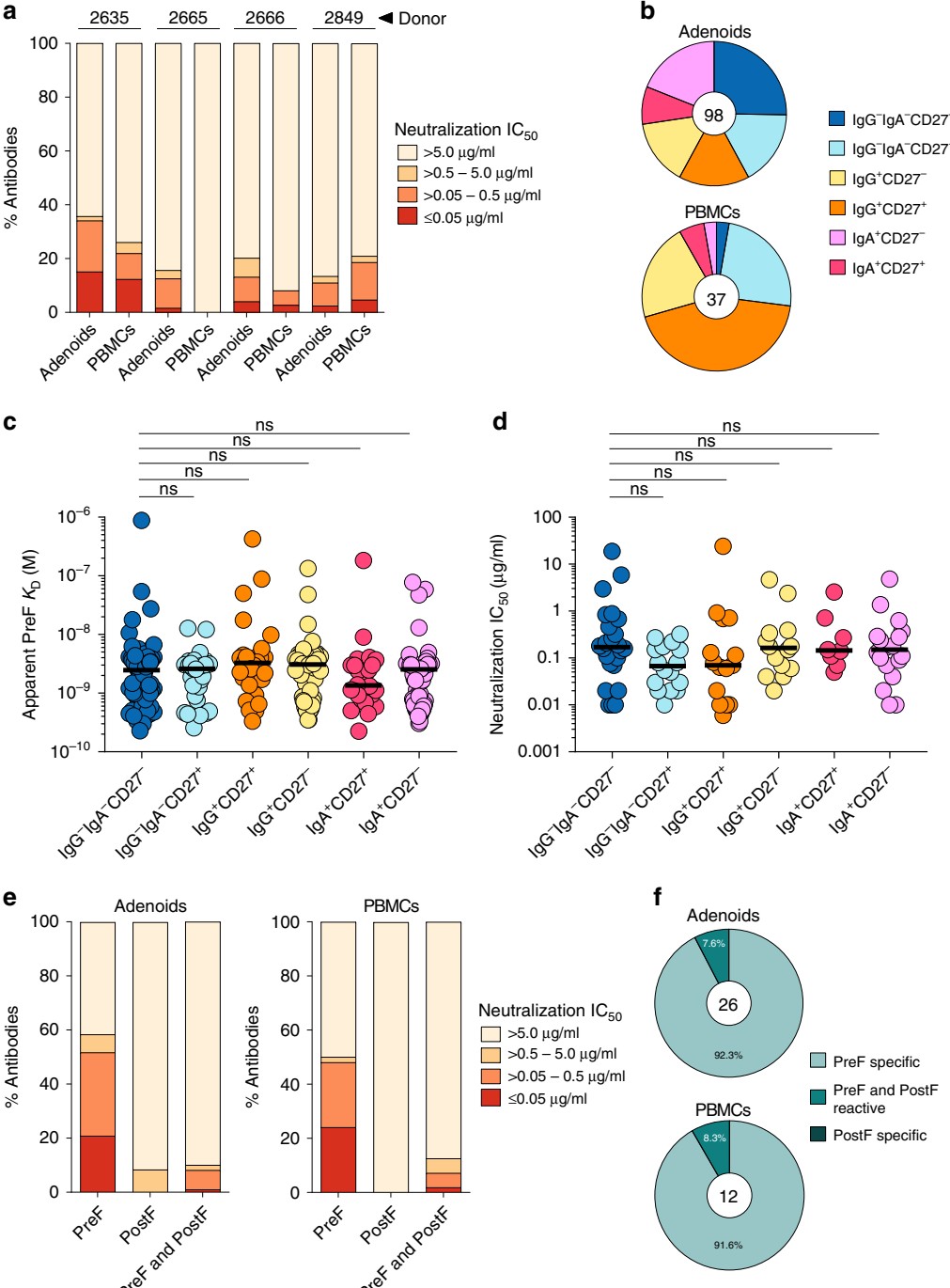

**Fig. 4** Neutralizing activity of RSV F-specific antibodies isolated from adenoids and PBMCs. **a** Percentage of antibodies isolated from adenoids and PBMCs with the indicated neutralization potencies ($n = 808$ antibodies). **b** Pie charts show the distribution of memory B cell subsets from which neutralizing antibodies were isolated. The number in the center of the pie denotes the number of neutralizing antibodies. **c** Apparent binding affinities of adenoid-derived antibodies isolated from each memory B cell subset. **d** Neutralizing activity of adenoid-derived antibodies isolated from each memory B cell subset. Only antibodies with detectable binding and neutralizing activity are plotted. **e** Percentage of preF-specific, preF/postF-cross-reactive, or postF-specific antibodies with the indicated neutralization potencies. Antibodies from all donors were pooled for this analysis. **f** Pie charts show the percentage of highly potent neutralizing antibodies (IC$_{50}$ ≤ 0.05 ug/ml) derived from either adenoids or PBMCs that are preF-specific, postF-specific, or preF/postF-cross-reactive. The number in the center of the pie denotes the number of highly potent neutralizing antibodies isolated from adenoids or PBMCs. Statistical comparisons were made using the Mann-Whitney test (n.s. = not significant). Results are representative of two independent experiments

In summary, our findings demonstrate that (i) adenoid tissue contains a higher proportion of high affinity neutralizing antibodies compared to peripheral blood, (ii) a relatively large fraction of RSV F-specific neutralizing antibodies in adenoid tissue originate from atypical memory B cells, and (iii) the majority of potent neutralizing antibodies isolated from both compartments target epitopes exclusively expressed on pre-F.

## Discussion

A detailed understanding of mucosal and systemic immune responses to natural RSV infection will facilitate the design and evaluation of RSV vaccine candidates. Although previous studies have shown that mucosal antibody responses are important for protection against RSV in both humans and animal models, the specificities and functional activities of these antibodies have remained undefined. In this study, we used a high-throughput B cell cloning platform to analyze and compare the memory B cell response to natural RSV infection in both adenoid and peripheral blood.

Interestingly, RSV F-specific B cell responses were observed in the adenoids of all 6 donors analyzed, whereas such responses were only detected in the peripheral blood samples of 4 of the 6 donors. In addition, in most donors studied, a larger proportion of adenoid-derived antibodies displayed high affinity binding and potent neutralizing activity compared to PBMC-derived antibodies. These results provide evidence that RSV-specific memory B cells are induced and maintained within adenoid tissue and suggest that this local response may be more robust and/or durable than the corresponding systemic response. Upon re-exposure to RSV, these memory B cells may undergo differentiation and expansion, with subsequent migration to effector tissues such as the tracheal lamina propria and lung. Hence, adenoidectomy may result in a reduction of local immune competence against RSV, as previously demonstrated by diminished poliovirus-specific antibody levels in nasal secretions from children following tonsillectomy and adenoidectomy[38]. Notably, one limitation of the binding and neutralization studies is that all of the antibodies were tested in an IgG1 format, regardless of the original isotype. Thus, the binding affinities and neutralization potencies may not be a true reflection of the binding or neutralization potentials of IgA or IgM antibodies due to the high avidity associated with these isotypes.

The adenoids of all donors studied contained a high frequency of RSV F-specific memory B cells that contained a high load of somatic mutations but were not isotype-switched and lacked expression of the classical memory B cell marker CD27. Although RSV F-specific B cells with this surface phenotype were also present in peripheral blood, the frequency was significantly lower than that observed in adenoid tissue and the majority of these B cells encoded antibodies that contained little to no SHM. Previous studies have also reported the presence of IgM$^+$IgD$^+$CD27$^-$ and IgM$^+$IgD$^-$CD27$^-$ memory B cells in peripheral blood with a similarly low frequency of SHM[39,40]. Notably, unlike previously described atypical memory B cell subsets in tonsillar tissue (IgG$^+$CD27$^-$FcRL4$^+$) or peripheral blood (IgG$^+$CD27$^-$)[29,41–45], the atypical memory B cell subset described here did not express FcRL4 or IgG and was heterogeneous with respect to IgM and IgD expression. Notably, 15–28% of these RSV F-specific CD27$^-$ IgM$^+$/IgD$^+$ B cells expressed CD45RB$^{MEM55}$, a glycosylation-dependent epitope that has been shown to be differentially expressed during human B cell development[36,46]. Importantly, previous studies have shown that marginal zone B cells in human gut associated lymphoid tissues, spleen, and tonsil are phenotypically linked to a CD27$^-$CD45RB$^{MEM55+}$ precursor population, which is enriched in young children compared to adults[47,48]. Hence, future studies should investigate whether this unique subset of adenoid-resident memory B cells is also present in adult donors.

Notably, we detected RSV F-specific IgA-committed memory B cells in both adenoid and peripheral blood for all donors, which is inconsistent with a previous report showing that IgA memory B cell responses are absent in peripheral blood after experimental RSV infection in healthy adult donors[19]. The reason(s) for this discrepancy are unclear but may be due to differences in the RSV-specific B cell response to experimental versus natural RSV infection or differences between responses in adults versus young children.

Previous studies have shown that RSV antibodies that bind to pre-F-specific surfaces are generally more potent than those that recognize epitopes expressed on both pre-F and post-F or only on post-F[5–7,13,14]. Correspondingly, we found that over 90% of the potent neutralizing antibodies isolated from both adenoid and peripheral blood recognized epitopes exclusively expressed on pre-F. The high abundance of pre-F-specific neutralizing antibodies and near absence of post-F-reactive neutralizing antibodies in adenoid tissue suggests that mucosal vaccines that preserve pre-F-specific antigenic surfaces may be required to induce high titers of protective antibodies. Although the majority of RSV mucosal vaccines in clinical development are particle-based or vector-based[4], it has been shown that pre-F can spontaneously trigger to adopt the post-F conformation on the viral surface[49], underscoring the importance of carefully evaluating the antigenic properties of such vaccine candidates. Notably, the extensive panel of antibodies described here could be used as reagents to measure the pre-fusion and post-fusion F content of these vaccines.

In conclusion, we have shown that adenoids serve as an induction site for RSV F-specific memory B cell responses and that a large proportion of this response is comprised of atypical IgM$^+$ and/or IgD$^+$ memory B cells. Future studies will be required to delineate the origin and function of these cells and to elucidate their role in protection against RSV and potentially other respiratory infections. Finally, the observation that the vast majority of adenoid-derived neutralizing antibodies target epitopes exclusively expressed on pre-F provides strong support for the development of pre-F-based mucosal vaccines that induce robust local responses.

## Methods

**Sample collection.** Study subjects aged 2.5–4 years of age were identified during their pre-operative visit to Otolaryngology, where an independent decision was made about the clinical indications for their tonsillectomy/adenoidectomy. Informed consent to participate in this study was obtained during the pre-operative visit. Heparinized blood (6–10 cc) was obtained from subjects at the time of surgery. Adenoid tissue removed during surgery was transferred to the laboratory for immediate processing. Blood and tissue samples were processed in the Immune Monitoring and Flow Cytometry core laboratory at the Geisel School of Medicine at Dartmouth to obtain plasma and to isolate adenoid-derived and peripheral blood-derived B cells. Isolated cells and plasma were stored frozen in aliquots. This study complies with all relevant ethical regulations for work with human participants and was approved by the Committee for the Protection of Human Subjects, Dartmouth-Hitchcock Medical Center and Dartmouth College.

**Production of RSV F sorting probes.** PreF (DS-Cav1) and postF (F ΔFP) trimers were produced with a single biotinylated C-terminal AviTag and then coupled to streptavidin-PE or streptavidin-APC[5]. Expression vectors containing each trimer with either a C-terminal 6x His-tag–AviTag or a C-terminal StrepTag II were co-transfected into FreeStyle 293-F cells at a 1:2 ratio. The protein was purified from the cell supernatant using Ni-nitrilotriacetic acid (NTA) resin to remove trimers lacking the 6× His-tag–AviTag and then purified over StrepTactin resin to remove those lacking the StrepTag II. Additonal wash steps removed trimers containing only one StrepTagII, and the remaining trimers containing two StrepTagII monomers and one 6× His-tag-AviTag monomer were then biotinylated using BirA biotin ligase (Avidity). The biotinylated proteins were separated from excess biotin by size-exclusion chromatography using a Superdex 200 column (GE Healthcare) in PBS. Dual-labeled tetramers were generated fresh for each experiment by incubating the trimers with premium-grade PE-labeled or APC-labeled streptavidin (Life Technologies) for 20 min on ice at a molar ratio of 4:1.

**Single B cell sorting.** Adenoid and PBMC samples were stained with anti-human APC-Cy7-conjugated CD19 (1:100, BioLegend #302218), APC-Cy7-conjugated CD20 (1:100, BioLegend #302314), PerCP-Cy5.5-conjugated CD3 (1:100, BioLegend #300430), PerCP-Cy5.5-conjugated CD8 (1:100, BioLegend #344710), PerCP-Cy5.5-conjugated CD14 (1:100, Thermo Fisher Scientific 45-0149-42), PerCP-Cy5.5-conjugated CD16 (1:100, BioLegend #302028), PECy7-conjugated FcRL4 (1:50, BioLegend #340208), BV605-conjugated IgG (1:100, BD Biosciences

#563246), 488-conjugated IgA (1:200, Abcam #ab98553), BV421-conjugated CD27 (1:100, BioLegend #356418), and a mixture of dual-labeled pre-F and post-F tetramers (25 nM each). To determine the percentage of RSV-F specific B cells expressing IgM, IgD, CD45RB, and CD5, the adenoid samples were stained with anti-human APC-Cy7-conjugated CD19, APC-Cy7-conjugated CD20, PerCP-Cy5.5-conjugated CD3, PerCP-Cy5.5-conjugated CD8, PerCP-Cy5.5-conjugated CD14, PerCP-Cy5.5-conjugated CD16, PE-Cy7-conjugated IgM (1:100, BioLegend #314532), BV510-conjugated IgD (1:100, BD Bdiosciences #563034), BV605-conjugated CD5 (1:100, BD Biosciences #563945), FITC-conjugated CD45RB (1:100, BioLegend #310205), BV421-conjugated CD27, and a mixture of dual-labeled pre-F and post-F tetramers (25 nM each). Tetramers were prepared fresh for each experiment, and B cells binding to the RSV F tetramers were single cell sorted. Single cells were sorted using a BD FACS Aria II (BD Biosciences) into 96-well PCR plates (BioRAD) containing 20 uL/well of lysis buffer [5 uL of 5× first strand cDNA buffer (Invitrogen), 0.625 uL of NP-40 (New England Biolabs), 0.25 uL RNaseOUT (Invitrogen), 1.25 uL dithiothreitol (Invitrogen), and 12.6 uL dH2O]. Plates were immediately stored at −80 °C.

**Amplification and cloning of antibody variable genes**. Antibody variable genes (VH and VL) were amplified by reverse transcription PCR and nested PCRs using cocktails of IgG-specific, IgA-specific, IgD-specific, and IgM-specific primers (Supplementary Data 2). The primers used in the second round of PCR contained 40 base pairs of 5′ and 3′′ homology to the digested expression vectors, which allowed for cloning by homologous recombination into S. cerevisiae. Yeast were transformed using the lithium acetate method of chemical transformation, in which yeast are incubated at 42 °C for 45 min with PEG 3350 (50% w/v), 1 M lithium acetate, boiled salmon sperm DNA (2 mg/mL), and the plasmid DNA[50]. Ten microliters of unpurified VH and VL PCR product and 200 ng of the digested expression vectors were used per transformation reaction. Following transformation, individual yeast colonies were picked for sequencing and production.

**Expression and purification of IgGs**. IgGs were expressed in S. cerevisiae cultures grown in 24-well plates. After 6 days, the culture supernatant was harvested by centrifugation and IgGs were purified by protein A-affinity chromatography[51]. The bound antibodies were eluted with 200 mM acetic acid/50 mM NaCl (pH 3.5) into 1/8th volume 2 M Hepes (pH 8.0), and buffer-exchanged into PBS (pH 7.0).

**Biolayer interferometry binding analysis**. IgG binding to pre-F (DS-Cav1) and post-F (F ΔFP) was measured by biolayer interferometry (BLI) using a FortéBio Octet HTX instrument (Pall Life Sciences). For high-throughput $K_D$ determination, IgGs were immobilized on anti-human IgG quantitation biosensors (Pall Life Sciences) and exposed to 100 nM antigen in PBS with 0.1% BSA (PBSF) for an association step, followed by a dissociation step in PBSF. Data were analyzed using the FortéBio Data Analysis Software 7. $K_D$ values were calculated for antibodies with BLI responses >0.1 nm, and the data were fit to a 1:1 binding model to calculate association and dissociation rate constants. $K_D$ values were calculated using the ratio kd/ka.

**Plasma neutralization assay**. Plasma samples were tested for RSV neutralization in microtiter assays using a recombinant RSV-expressing Renilla luciferase (rA2-Rluc)[37]. Hep2 cells were deposited into 96-well plates at a density of $1.8 \times 10^4$ cells per well in 100 uL of minimal essential media (MEM) with 2% FBS/1X penicillin-streptomycin solution (2% MEM) and allowed to adhere overnight at 37 °C. On the day of the assay, plasma samples were serially diluted 2-fold (1:200 to 1:128,000) in 2% MEM containing rA2-Rluc and incubated for 30 min at 37 °C. Culture media was aspirated from the Hep2 cells followed by the addition of 100 uL per well of the plasma/rA2-Rluc mixture to duplicate wells. Cultures were maintained at 37 °C for 24 h and luciferase expression was quantified in cell lysates using the Renilla Luciferase Assay System (E2820, Promega, Madison, WI). Relative light units (RLU) were measured on a BioTek Synergy 2 microplate reader. Neutralization is expressed as the reciprocal of the highest plasma dilution to yield a 50% reduction in RLU as compared to control wells with no added plasma. The Hep2 cell line was obtained from ATCC.

**Adenoid neutralization assay**. Adenoid tissue collected on the day of surgery was placed in a sterile 10 cm culture dish. A 1.8 cm circular disc of soft absorbent filter paper (Leukosorb #BSP0669, Pall Corporation, Port Washington NY) was applied to the mucosal surface of the tissue. One ml of PBS with added protease inhibitors (PI; Bestatin 0.1 ug/ml; Aprotinin 1 ug/ml; AEBSF 0.5 ug/ml; Leupeptin 5 ug/ml; Millipore Sigma, St. Louis MO) was added directly to the tissue to moisten the disc. The tissue was allowed to stand for 30 min at room temperature. Excess PBS + PI was then pipetted from the tissue into a 15 ml conical tube. The filter paper disc was collected with sterile forceps and placed into a separate 15 ml tube. An additional 0.5 ml of PBS + PI was added, and the tube was centrifuged at $1900 \times g$ for 10 min. Supernatant recovered directly from the tissue and from the filter disc was retained and tested for RSV neutralizing activity. Supernatants were serially diluted 2-fold (1:4 to 1:256) and tested using the rA2-Rluc microtiter assay. Data is expressed as the dilution corresponding to a 50% inhibitory concentration (IC50)

compared to control wells with rA2-Rluc alone. The Hep2 cell line was obtained from ATCC.

**Reporting summary**. Further information on experimental design is available in the Nature Research Reporting Summary linked to this article.

## Data availability
The data that support the findings of this study are available from the corresponding author upon reasonable request.

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

## Acknowledgements

We thank J. McLellan and M. Gilman for providing the RSV preF and postF proteins. We also thank C. Williams, A. Wec and S.M. Eagol for assistance with figure preparation. PBMC processing was carried out in the Dartmouth Immune Monitoring and Flow Cytometry Core lab.

## Author contributions

L.M.W. and P.F.W. conceived the idea and supervised the research. L.M.W., P.F.W, E.C. and R.I.C. designed the experiments. W.F.W, L.S., D.P.M., E.C., R.I.C. and L.M.W. performed the experiments and analyzed the data. The manuscript was written by L.M.W. and L.S. with input from P.F.W., D.P.M. and R.I.C.

## Additional information

**Competing interests:** L.M.W., L.S. and D.P.M. are shareholders in Adimab, LLC. The remaining authors declare no competing interests.

