## [Peer Review File · Nature Communications]

Reviewers' Comments:

Reviewer #1:

Remarks to the Author:

The manuscript entitled "Respiratory Syncytial virus infection" induces potent mucosal B cell responses in young children" studies B-cell and antibody responses in local tissue adenoids associated with mucosal respiratory responses and contrast them with those that are found in the blood. The authors conclude that the adenoid derived antibodies showed overall higher binding affinities, translating to better neutralization potencies than those found in the blood. About a quarter of these antibodies are encoded by a unique B-cell population lacking class switching but having somatic hypermutations thus classified as B-memory cells even though they lack the classic memory markers like CD27.

Overall the body of work is of significant importance to the readership as it studies immune responses in the mucosal compartment thus having implication for vaccine design in terms of antigen selection and routes of immunization as well as adjuvant selection to get optimal mucosal responses. There are several major points that need to be clarified or caveats discussed for the manuscript to be acceptable for publication.

1) 25% of the adenoid antibodies are produced by IgM+ and/or IgD+ B cells therefore a more detailed FACS based phenotyping could have been done to better characterize this population e.g. what is the expression of CD5 on these cells. The tonsils are known to have a vast majority of the cells belonging to the marginal B-cell population. Alternatively, immunohistochemistry could be performed to locate these antigen specific cells in distinct anatomical site in the adenoid tissues. If none of the adenoid samples are available than the caveats for these B cell characterization needs to be discussed in the manuscript.

2) The authors express all their antibodies in IgG1 formats which may not be the true reflection of binding or neutralization potentials of IgM and IgA antibodies due to high avidity and differential effector functions associated with these isotypes (IgM and IgA). It would be interesting to use mono- or bi-valent Fabs in their Kd and neutralization determinations to compare with Fabs derived from the IgG and peripheral antibodies.

3) A more detailed analysis of somatic hypermutations found in the H and L-chains of these antibodies is warranted. This analysis would include the determination of Replacement (R) to Silent (S) ratio's of amino acid changes due to hypermutations in the CDR's and compare it to framework regions. Higher R/S ratios in the CDR's compared to framework is another hallmark of affinity maturation and selection in germinal centers/memory B cells.

4) Minor point: Poly reactivity of the antibodies in the IgG format is artificial as IgM is a pentamer and IgA a dimer. Therefore, multivalency of IgM and IgA leads to higher avidity and higher polyreactivity in vivo. This section may not be as relevant to the main message of the manuscript and could be removed from the manuscript.

Kalpit A Vora

Reviewer #2:

Remarks to the Author:

In this interesting study the authors have analysed B cells that bind RSV F fusion protein in preF and postF conformations. B cells studied were isolated from blood and adenoids of children who had not had clinical RSV infections. The assumption is that children would have encountered virus and this is supported by plasma neutralisation titres. The authors generate monoclonal antibodies from single RSV binding cells and characterise the specificity and neutralising capacity of antibodies derived from B cells with different phenotypes. Despite the relatively low number of tissue donors studied and inconsistencies between individuals, the data show some interesting features including the production of antibodies by B cells with the phenotype CD27-IgG-IgA- that have somatic mutation indicating that

they are not naïve. They further identify cells with the phenotype CD27-IgM+IgD+/- that bind RSV F by flow cytometry.

The authors should illustrate the appearance of the plots such as that in Figure 1 and Figure S1 over a range of values. Currently only the upper end of the range is illustrated.

The presence of somatic mutations in cells that appear otherwise (phenotypically) naive is interesting and the status of these cells as memory cells should be demonstrated in other ways. For example, the IGHV genes of the cells that bind RSV F that have the phenotype CD27-IgM+IgD+ could be sequenced and the sequences compared with the naive B cells from the same donors. These cells could be phenotyped in more detail, perhaps including alternative markers of memory B cells such as CD45RB clone MEM55.

The authors should illustrate the B cell clone that spans the blood and adenoids in more detail. The DNA sequence should be illustrated rather than amino acids. This is relevant because the ES sequence illustrated as a mutation here is also seen in a polymorphic variant of VH4-34 at that place. The total DNA sequence could be informative and all of the mutations would be interesting. It would be helpful to see the CDR3 sequences in the same alignment.

The gating of FcRL4+ cells in blood and adenoids should be illustrated. Positive cells in adenoids and blood might be expected to be quite different.

The authors should refer to the origin of the cells they have studied as adenoids and blood and not classify them as mucosal and systemic. There are problems with both terms 'mucosal' and 'systemic' in the ways they are used. Using 'mucosal' wrongly attributes the features they describe to a larger group lymphoid tissues than the adenoids. The blood contains cells with nodal and splenic circulations and origins as well as mucosal cells and should therefore be referred to as 'blood' and the cells as PBMC, and not 'systemic'.

CD27-IgM+IgD- B cells with somatic mutations have been described previously, and this should be cited. <https://www.ncbi.nlm.nih.gov/pubmed/22566870>.

An association between CD27- memory and mucosal surfaces has previously been proposed and references should be cited, though admittedly these were not IgM's.

<https://www.ncbi.nlm.nih.gov/pubmed/21690558> and

<https://www.ncbi.nlm.nih.gov/pubmed/26150533>

Although the authors investigate polyspecificity, they do not consider cross reactivity with/ binding of microbiota/ bacterial antigens. This is highly relevant in the adenoids.

Reviewer #1:

1) 25% of the adenoid antibodies are produced by IgM⁺ and/or IgD⁺ B cells therefore a more detailed FACS based phenotyping could have been done to better characterize this population e.g. what is the expression of CD5 on these cells. The tonsils are known to have a vast majority of the cells belonging to the marginal B-cell population. Alternatively, immunohistochemistry could be performed to locate these antigen specific cells in distinct anatomical site in the adenoid tissues. If none of the adenoid samples are available than the caveats for these B cell characterization needs to be discussed in the manuscript.

We thank the reviewer for this suggestion, and we have now analyzed this B cell population for expression of both CD5 and CD45RB. This analysis revealed that the vast majority (85-95%) of RSV F-specific atypical memory B cells (defined as CD19⁺ CD27⁺ IgM⁺ and/or IgD⁺) either lacked expression of both CD5 and CD45RB or were CD5⁻ CD45RB⁺. This data is now included in Fig. 2D and Supplementary Figure 8.

2) The authors express all their antibodies in IgG1 formats which may not be the true reflection of binding or neutralization potentials of IgM and IgA antibodies due to high avidity and differential effector functions associated with these isotypes (IgM and IgA). It would be interesting to use mono- or bi-valent Fabs in their Kd and neutralization determinations to compare with Fabs derived from the IgG and peripheral antibodies.

We agree with the reviewer that it would be interesting to compare the neutralization potencies of the Fab fragments, IgAs, and IgMs with that observed for the IgGs. However, generating Fab fragments for over 800 antibodies, or re-cloning and expressing these mAbs as IgAs or IgMs, would be an enormous technical hurdle. Therefore, we have discussed this caveat in the discussion section.

3) A more detailed analysis of somatic hypermutations found in the H and L-chains of these antibodies is warranted. This analysis would include the determination of Replacement (R) to Silent (S) ratio's of amino acid changes due to hypermutations in the CDR's and compare it to framework regions. Higher R/S ratios in the CDR's compared to framework is another hallmark of affinity maturation and selection in germinal centers/memory B cells.

We thank the reviewer for this suggestion and have now included this data in supplementary figure 7. Indeed, the results show that the antibodies derived from this atypical memory B cell population have a higher R/S ratio in the CDRs compared to the framework regions, providing further evidence for antigen-mediated selection.

4) Minor point: Poly reactivity of the antibodies in the IgG format is artificial as IgM is a pentamer and IgA a dimer. Therefore, multivalency of IgM and IgA leads to higher avidity and higher polyreactivity in vivo. This section may not be as relevant to the main message of the manuscript and could be removed from the manuscript.

We thank the reviewer for raising this important point and have now removed the polyreactivity data from the manuscript

Reviewer #2:

1) The authors should illustrate the appearance of the plots such as that in Figure 1 and Figure S1 over a range of values. Currently only the upper end of the range is illustrated.

We have now included the flow cytometry plots for all donors in supplementary figure 1.

2) The presence of somatic mutations in cells that appear otherwise (phenotypically) naive is interesting and the status of these cells as memory cells should be demonstrated in other ways. For example, the IGHV genes of the cells that bind RSV F that have the phenotype CD27⁻IgM⁺IgD⁺ could be sequenced and the sequences compared with the naive B cells from the same donors. These cells could be phenotyped in more detail, perhaps including alternative markers of memory B cells such as CD45RB clone MEM55.

We thank the reviewer for this suggestion, and we have now analyzed this B cell population for expression of both CD5 and CD45RB (clone MEM55). This analysis revealed that the vast majority (85-95%) of RSV F-specific atypical memory B cells (defined as CD19⁺CD27⁻IgM⁺ and/or IgD⁺) either lacked expression of both CD5 and CD45RB or were CD5⁻CD45RB⁺. This data is now included in Fig. 2D and Supplementary Figure 8. We have also performed a more detailed analysis of somatic mutations in these antibodies by determining the replacement to silent ratios (R/S ratio) of amino acid changes in the CDRs and compared it to the R/S ratio in the framework regions. This analysis shows that these antibodies have higher R/S ratios in the CDRs compared to framework regions, which is a hallmark of affinity maturation and selection in germinal centers.

3) The authors should illustrate the B cell clone that spans the blood and adenoids in more detail. The DNA sequence should be illustrated rather than amino acids. This is relevant because the ES sequence illustrated as a mutation here is also seen in a polymorphic variant of VH4-34 at that place. The total DNA sequence could be informative and all of the mutations would be interesting. It would be helpful to see the CDR3 sequences in the same alignment.

We agree that this would be more informative and have now included both the DNA and protein alignments in Figure S5.

4) The gating of FcRL4+ cells in blood and adenoids should be illustrated. Positive cells in adenoids and blood might be expected to be quite different.

We have now included a supplementary figure showing the gating of FcRL4+ cells in blood and adenoids. These results show that, in 3 out of 4 donors, the proportion of B cells expressing FcRL4 is higher in adenoids (2-10x) compared to peripheral blood.

5) The authors should refer to the origin of the cells they have studied as adenoids and blood and not classify them as mucosal and systemic. There are problems with both terms 'mucosal' and 'systemic' in the ways they are used. Using 'mucosal' wrongly attributes the features they describe to a larger group lymphoid tissues than the adenoids. The blood contains cells with nodal and splenic circulations and origins as well as mucosal cells and should therefore be referred to as 'blood' and the cells as PBMC, and not 'systemic'.

We agree that this would be clearer, and we have now modified the text accordingly.

6) CD27-IgM+IgD- B cells with somatic mutations have been described previously, and this should be cited. <https://www.ncbi.nlm.nih.gov/pubmed/22566870>. An association between CD27- memory and mucosal surfaces has previously been proposed and references should be cited, though admittedly these were not IgM's. <https://www.ncbi.nlm.nih.gov/pubmed/21690558> and <https://www.ncbi.nlm.nih.gov/pubmed/26150533>

We thank the reviewer for pointing this out and we have now referenced these studies in the discussion section.

7) Although the authors investigate polyspecificity, they do not consider cross reactivity with/ binding of microbiota/ bacterial antigens. This is highly relevant in the adenoids.

Based on the comments raised by Reviewer #1, we have decided to remove the polyreactivity data from the manuscript.

We hope that the modifications to the manuscript meet your approval.

Reviewers' Comments:

Reviewer #1:

Remarks to the Author:

The revised manuscript has been improved and reads more balanced with the caveats of data clearly discussed. It would still be interesting to take a few representative (n=3-5) IgM or IgA antibodies from adenoids and compare their binding and neutralization potencies in a Fab format to the IgG's found in the blood.

Reviewer #2:

Remarks to the Author:

Some of my concerns have been addressed but there are still some outstanding issues.

1. The title should make it clear that this is a study of adenoids and should not refer to 'potent mucosal B cells responses'. '... B cell responses in the adenoids of young children' would be more appropriate

2. The manuscript mentions a clone of B cells that spans the blood and adenoids. In my opinion the sequence alignment provided does not support this. Whilst the heavy chain sequences that have been aligned use the same relatively common V and J segments and have the same CDR3 length, the sequence homology in the CDR3 is poor. In the alignment of the junctional region (copied below) there are 9 mismatches in a sequence of 17 nt. If these and the flanking sequences are put into IMGT V quest the CDR3 D segment alignment is different. It would be unlikely that all of these differences are somatic hypermutations given the overall mutation profile of the heavy and light chains.

```
tg ggt ggg aac tcg ggg  
tt ggg ggt cgc agt gcc
```

In general there is less variability at kappa compared to heavy chain junctions. In this example there is a mismatched codon in the J sequence.

On balance the claim that this is a clone is not convincing and all reference to it and the discussion of it should be removed from the manuscript.

3. From figure 2E it appears that up to approximately 1/3 of CD27-IgM+IgD+ B cells recognising RSV express CD45RB. The text on page 8 says that 'between 85-95% of these IgM+ and/or IgD+ B cells either lacked expression of both CD5 and CD45RB or were CD5-CD45RB+'. Presumably the 85-95% refers to 'lacked expression of both CD5 and CD45RB'? The fact that quite a few express CD45RB should be included and a range for CD45RB+ included in addition to the 85-95% figure. This CD27-IgM+IgD+CD45RB+ phenotype is shared by memory B cells and the marginal zone precursor population that is enriched in children and is therefore highly relevant to this manuscript and should be discussed. References for this include

<https://www.ncbi.nlm.nih.gov/pubmed/21278234>

<https://www.ncbi.nlm.nih.gov/pubmed/24211716>

<https://www.ncbi.nlm.nih.gov/pubmed/30242242>

Reviewer #1:

- 1) The revised manuscript has been improved and reads more balanced with the caveats of data clearly discussed. It would still be interesting to take a few representative (n=3-5) IgM or IgA antibodies from adenoids and compare their binding and neutralization potencies in a Fab format to the IgG's found in the blood.

We agree with the reviewer that this would be an interesting experiment. However, since the antibodies were all expressed in the same backbone (IgG1), we feel that the current experiments provide a sufficient comparison of the relative neutralization potencies and binding affinities of antibodies that were originally different isotypes. Furthermore, previous studies have shown that most RSV antibodies neutralize equally well as IgG or Fab fragments(1), presumably due to the relatively low density of RSV prefusion F spikes on the viral surface. However, we have discussed the caveat associated with using IgGs versus IgAs/IgMs in the discussion section.

Reviewer #2:

- 1) The title should make it clear that this is a study of adenoids and should not refer to 'potent mucosal B cells responses'. '... B cell responses in the adenoids of young children' would be more appropriate.

We thank the reviewer for this suggestion and have now modified the title accordingly.

- 2) The manuscript mentions a clone of B cells that spans the blood and adenoids. In my opinion the sequence alignment provided does not support this. Whilst the heavy chain sequences that have been aligned use the same relatively common V and J segments and have the same CDR3 length, the sequence homology in the CDR3 is poor. In the alignment of the junctional region (copied below) there are 9 mismatches in a sequence of 17 nt. If these and the flanking sequences are put into IMGT V quest the CDR3 D segment alignment is different. It would be unlikely that all of these differences are somatic hypermutations given the overall mutation profile of the heavy and light chains.

```
tg ggt ggg aac tcg ggg
tt ggg ggt cgc agt gcc
```

In general there is less variability at kappa compared to heavy chain junctions. In this example there is a mismatched codon in the J sequence.

We thank the reviewer for pointing this out and agree that this may be an example of convergent sequences as opposed to somatic variation, and we have now removed this data and discussion from the manuscript.

- 3) From figure 2E it appears that up to approximately 1/3 of CD27-IgM+IgD+ B cells recognising RSV express CD45RB. The text on page 8 says that 'between 85-95% of these IgM+ and/or IgD+ B cells either lacked expression of both CD5 and CD45RB or were CD5- CD45RB+'. Presumably the 85-95% refers to 'lacked expression of both CD5 and CD45RB'? The fact that quite a few express CD45RB should be included and a range for CD45RB+ included in addition to the 85-95% figure. This CD27-IgM+IgD+CD45RB+ phenotype is shared by memory B cells and the marginal zone precursor population that is enriched in children and is therefore highly relevant to this manuscript and should be discussed. References for this include
<https://www.ncbi.nlm.nih.gov/pubmed/21278234>
<https://www.ncbi.nlm.nih.gov/pubmed/24211716>
<https://www.ncbi.nlm.nih.gov/pubmed/30242242>

We apologize for the lack of clarity in this section. We have now re-worded this paragraph to make it clear that the majority of CD27-IgM+IgD+ B cells recognizing RSV are either CD5—CD45RB MEM55— (62-83%) or CD5—CD45RBMEM55+(15-28%). We have also emphasized in the text that a relatively large fraction of these cells expresses CD45RBMEM55+and discussed previous studies showing that these cells represent a marginal zone precursor population that is enriched in children compared to adult donors.

We hope that the modifications to the manuscript meet your approval.